Homology modeling and docking studies of a Δ9-fatty acid desaturase from a Cold-tolerant Pseudomonas sp. AMS8

Garba Lawal 1 2 3
Mohamad Yussoff Mohamad Ariff 4
Abd Halim Khairul Bariyyah 4
Ishak Siti Nor Hasmah 1
Mohamad Ali Mohd Shukuri 1 5
Oslan Siti Nurbaya 1 5
Raja Abd. Rahman Raja Noor Zaliha rnzaliha@upm.edu.my 1 2
1 Enzyme and Microbial Technology Research Centre, Faculty of Biotechnology and Biomolecular Sciences, Universiti Putra Malaysia , Serdang , Selangor , Malaysia
2 Department of Microbiology, Faculty of Biotechnology and Biomolecular Sciences, Universiti Putra Malaysia , Serdang , Selangor , Malaysia
3 Department of Microbiology, Faculty of Science, Gombe State University , Gombe , Gombe State , Nigeria
4 Department of Biotechnology, Kulliyyah of Science, International Islamic University Malaysia , Kuantan , Pahang Darul Makmur , Malaysia
5 Department of Biochemistry, Faculty of Biotechnology and Biomolecular Sciences, Universiti Putra Malaysia , Serdang , Selangor , Malaysia
Silva Pedro
Electronic publication date: 2018 Mar 19
Publication date: 2018
Volume: 6
Electronic Location ID: e4347
Received 2017 May 8; Accepted 2018 Jan 19
Copyright: ©2018 Garba et al.
Copyright year: 2018
Copyright holder: Garba et al.
License: This is an open access article distributed under the terms of the Creative Commons Attribution License, which permits unrestricted use, distribution, reproduction and adaptation in any medium and for any purpose provided that it is properly attributed. For attribution, the original author(s), title, publication source (PeerJ) and either DOI or URL of the article must be cited.
License URL: https://creativecommons.org/licenses/by/4.0/

Keywords: Cold-tolerant Pseudomonas sp. AMS8, Molecular docking, Homology modeling, δ9-fatty acid desaturase, Palmitic acid

Funding: Universiti Putra Malaysia GP-IPS/2016/9471000 FRGS2015-207-0448 FRGS2015-208-0449 The research was funded by the Putra grant, Universiti Putra Malaysia (GP-IPS/2016/9471000), FRGS2015-207-0448 and FRGS2015-208-0449. The funders had no role in study design, data collection and analysis, decision to publish, or preparation of the manuscript.

==============================
Membrane-bound fatty acid desaturases perform oxygenated desaturation reactions to insert double bonds within fatty acyl chains in regioselective and stereoselective manners. The Δ9-fatty acid desaturase strictly creates the first double bond between C9 and 10 positions of most saturated substrates. As the three-dimensional structures of the bacterial membrane fatty acid desaturases are not available, relevant information about the enzymes are derived from their amino acid sequences, site-directed mutagenesis and domain swapping in similar membrane-bound desaturases. The cold-tolerant Pseudomonas sp. AMS8 was found to produce high amount of monounsaturated fatty acids at low temperature. Subsequently, an active Δ9-fatty acid desaturase was isolated and functionally expressed in Escherichia coli. In this paper we report homology modeling and docking studies of a Δ9-fatty acid desaturase from a Cold-tolerant Pseudomonas sp. AMS8 for the first time to the best of our knowledge. Three dimensional structure of the enzyme was built using MODELLER version 9.18 using a suitable template. The protein model contained the three conserved-histidine residues typical for all membrane-bound desaturase catalytic activity. The structure was subjected to energy minimization and checked for correctness using Ramachandran plots and ERRAT, which showed a good quality model of 91.6 and 65.0%, respectively. The protein model was used to preform MD simulation and docking of palmitic acid using CHARMM36 force field in GROMACS Version 5 and Autodock tool Version 4.2, respectively. The docking simulation with the lowest binding energy, −6.8 kcal/mol had a number of residues in close contact with the docked palmitic acid namely, Ile26, Tyr95, Val179, Gly180, Pro64, Glu203, His34, His206, His71, Arg182, Thr85, Lys98 and His177. Interestingly, among the binding residues are His34, His71 and His206 from the first, second, and third conserved histidine motif, respectively, which constitute the active site of the enzyme. The results obtained are in compliance with the in vivo activity of the Δ9-fatty acid desaturase on the membrane phospholipids.

Introduction

Fatty acid desaturase enzymes perform desaturation reactions which strictly create a double bond within fatty acyl chain in regioselective and stereoselective manners. Phylogenetically, the enzymes have been broadly divided into two unrelated classes as the acyl-acyl carrier protein and membrane-bound fatty acid desaturases. The class of the acyl-acyl carrier proteins specifically catalyses the production of oleic acid (C18:1) from stearic acid (C18:0) in plants whereas that of the membrane-bound desaturases represent the most widely distributed form of the enzymes predominantly found in bacteria and eukaryotes (Hashimoto et al., 2008; Kachroo et al., 2007).

In the mechanism of oxygen-dependent desaturation reactions, the fatty acid desaturases activate molecular oxygen using their active-site diiron centre which is shared by several proteins such as ribonucleotide reductase, methane monooxygenase, rubrerythrins, and a range of oxidase enzymes. Relevant information about the tuning of the diiron centers in relation to various chemical reactivity have been made available through comparisons of the diiron clusters of many diiron-containing enzymes (Sazinsky & Lippard, 2006; Shanklin et al., 2009; Yoon & Lippard, 2004). Nevertheless, disparities in various protein to protein interactions, amino acid sequence and reaction outcomes confound the analysis. Research on fatty acid desaturases and similar enzymes created an avenue to conducting structure-function analyses due to a wide range of reactions performed on like substrates by the close homologous enzymes (Lee et al., 1998; Shanklin & Cahoon, 1998; Shanklin et al., 2009).

The amino acid sequences of both the integral membrane desaturases in (bacteria and eukaryotes) and acyl-acyl carrier protein desaturases of plants contain conserved histidine boxes predicted as the essential catalytic sites of the enzymes (Alonso et al., 2003). However, the former enzymes contained three conserved-histidine motifs labelled as ‘HXXXXH’, ‘HXXHH’ and ‘HXXHH’ whereas the latter contained twice conserved-histidine motifs as EXXH (Alonso et al., 2003; Lindqvist et al., 1996). As the three-dimensional structures of the bacterial membrane fatty acid desaturases are still unavailable, relevant information about the enzymes are derived from the amino acid sequences, site-directed mutagenesis, and domain swapping in similar membrane-bound desaturases coupled with homology modelling (Venegas-Calerón et al., 2006). The cold-tolerant Pseudomonas sp. AMS8 was able to produce high amount of monounsaturated fatty acids at 4 °C (Garba et al., 2016). Subsequently, an active Δ9-fatty acid desaturase was isolated and functionally expressed in Escherichia coli. The enzyme was found to catalyse conversion from membrane associated palmitic to palmitoleic acid (Garba et al., 2016a). In this paper we report homology modeling and docking studies of Δ9-fatty acid desaturase from a cold-tolerant Pseudomonas sp. AMS8 with palmitic acid as a substrate for the first time to the best of our knowledge.

Materials and Methods

Templates selection

BLASTP of the target protein was performed at the NCBI (https://blast.ncbi.nlm.nih.gov/Blast.cgi?PROGRAM=blastp) which showed 24 and 23% similarities to human integral membrane stearoyl-CoA desaturase (PDB ID: 4ZYO) and mouse stearoyl-coA desaturase (PDB ID: 4YMK), respectively. However, the mouse stearoyl-coA desaturase (PDB ID: 4YMK) was chosen as a template (based on its high resolution) to predict the three dimensional structure of the Δ9-fatty acid desaturase.

Structure prediction

The primary sequences of human (Uniprot ID: O00767) and mouse (Uniprot ID: P13516) desaturases were obtained from Uniprot protein databases. Moreover, the protein sequences of Δ9-fatty acid desaturases from several other Pseudomonas species were obtained from the GenBank. The transmembrane (TM) spanning region of Pseudomonas sp. AMS8 Δ9-fatty acid desaturase was predicted using a TM domain topology prediction program, CCTOP (http://cctop.enzim.ttk.mta.hu/). The CCTOP predicted the TM domains of the protein sequence based on the consensus of ten (10) different methods (Dobson, Reményi & Tusnády, 2015a; Dobson, Reményi & Tusnády, 2015b). The TM domains were modelled together with the remaining amino acid residues toward the C-terminus of the protein using MODELLER version 9.18 (Webb & Sali, 2014). Alignment input used in the MODELLER was derived from the pairwise alignments of both the template and model primary sequences using membrane proteins alignment tool (Stamm et al., 2013) whereas the secondary structure was predicted using PSIPRED tool (Buchan et al., 2013) and used as a guide to modelling the C-terminal domain.

Energy minimization and quality verification

The protein model generated by the MODELLER was ranked and scored using discrete optimised protein energy (DOPE) score. The top three models out of 50 models with the lowest DOPE scores were chosen and assessed using ERRAT and RAMPAGE server. The best model was selected for energy minimization to remove clashes between side chains using GROMACS and subsequently used in docking and molecular dynamics simulations. Further structural assessment was performed by simulating the homology models inside a membrane bilayer.

Active site prediction

Prior to docking simulations, the binding site for the Pseudomonas sp. AMS8 model was predicted using COACH (Buchan et al., 2013). The predicted active site was found in the vicinity of histidine rich region, which served as an already established potential binding site of the enzyme (this site was used for a targeted docking). Blind docking was also performed to bind palmitate on both the template and model structures.

Docking studies

Three dimensional (3D) structure of palmitic acid was obtained from the Pubchem (https://pubchem.ncbi.nlm.nih.gov). The energy minimized model of the Δ9-fatty acid desaturase and the palmitic acid (ligand) were prepared and used for molecular docking of the substrate onto the target proteins using Autodock tool Version 4.2 (Trott & Olson, 2010). Blind docking of the palmitate onto the modelled structure was performed using a pre-set simulation grid box size of 126 × 126 × 126 Å along the X, Y and Z axes and centred at 39.946, 40.191, 45.879 whereas the targeted docking grid box size was set to 70 × 70 × 60 Å dimension and centred at 43.946, 40.191, 33.879 of X, Y and Z coordinate, respectively. The docking simulations were performed for 100 runs using Lamarckian Genetic Algorithm (LGA). The results were evaluated using RMSD values, ligand-protein interactions, binding energy (ΔGbind) as well as a number of conformations existed in a populated cluster. The charge values of metal ions have been indicated to play a crucial role in predicting correct docking simulations. Previous report showed that different charge values of magnesium ions were verified and found a charge of +2 was too high, causing it to diminish due to partial binding of the adjacent groups (Chen et al., 2007). A docking study conducted on dioxyhypusine hydroylase (DOHH) with its substrate in the presence of Fe-Fe confirmed that Fe charge of +1 produced substrate binding mode which sufficiently agreed with the experimental data (Han et al., 2015). Thus, in this study, iron charges of 0 (default charge in Autodock), +1 and +2 were tested in the docking simulations for comparison. The ligand-protein interaction was visualized using Pymol (Trott & Olson, 2010) and VMD (Humphrey, Dalke & Schulten, 1996).

MD simulations

The model structure of the Δ9-fatty acid desaturase was simulated in an embedded 1-palmitoyl-2-oleoyl-sn-glycero-3-phosphocholine (POPC) bilayer. The protein-bilayer system was constructed using CHARMM-GUI Membrane builder (Jo et al., 2008). The atomistic MD simulations were performed using CHARMM36 force field (Huang & MacKerell, 2013) in GROMACS Version 5 (Pronk et al., 2013) within an integration time step of 20 fs. The simulation temperature was kept constant at 310 K by coupling the system to a heat bath using a Nose–Hoover thermostat with τT = 1 ps. Pressure was maintained at 1 atm using a Parinello-Rahman barostat and semiisotropic pressure with τP = 5 ps and a compressibility of 4.5e−5 bar −1. Long-range electrostatics was treated using particle mesh Ewald method with a cutoff of 12 Å. The 12 Å cutoff distance was used for van der Waals interactions. The systems were equilibrated for 1 ns restraining the C α atoms, followed by production runs of 50 ns each in triplicates. The data was analysed using GROMACS tools and VMD.

Results

Sequence of the Δ9-fatty acid desaturase protein and template identification

The Δ9-fatty acid desaturase was isolated from a cold-tolerant Pseudomonas sp. AMS8 and functionally expressed in Escherichia coli as confirmed by GCMS analysis which showed an active enzyme capable of increasing the overall palmitoleic acid content of the recombinant E. coli. Based on the GCMS analysis, a profound increase of the amount of palmitoleic acid from 10.5 to 21% was observed at 20 °C (Garba et al., 2016a). The protein had a molecular weight of 45 kDa and 394 amino acids which was already deposited at NCBI (accession number: AMX81567). Multiple sequences alignments of the template, the cold-tolerant Pseudomonas sp. AMS8 Δ9-fatty acid desaturase and sequences from several other Δ9-fatty acid desaturases have revealed the three conserved-histidine boxes common to all membrane-bound desaturases in bacteria (Garba et al., 2016b; Li et al., 2009), fungi (Chen et al., 2013) and animals (Bai et al., 2015) (Fig. 1). The human integral membrane stearoyl-CoA desaturase (PDB ID: 4ZYO) and mouse stearoyl-coA desaturase (PDB ID: 4YMK) have been solved to a resolution of 3.25 Å and 2.6 Å, respectively. The two structures share moderate sequence identities of 24 and 23% with the Pseudomonas sp. AMS8, respectively. However, the mouse stearoyl-coA desaturase was chosen as the template based on its higher resolution.

Figure 1 Multiple sequences alignments of protein sequence from Pseudomonas sp. AMS8 Δ9-fatty acid (AMX81567) and sequences from other desaturase proteins.

The transmembrane domains of the protein are indicated by letters A, B and C whereas the three conserved-histidine boxes common to all membrane-bound desaturases are shown by D, E, and F. The end of the model structure is at Arg252 indicated by an asterisk (*).

Model of the Δ9-fatty acid desaturase

To correctly model the TM domain of a membrane protein, it is necessary to appropriately predict its TM spanning region. CCTOP used 11 TM prediction programs including some of the best TM domain predictor such as TMHMM and HMMTOP to predict the TM domain of the target protein. Most of the prediction programs predicted that the target protein has three TM spanning regions as detailed in Fig. 2A, which gave a consensus domains of TM1 (Leu13–Leu33), TM2 (Leu135–Ile159) and TM3 (Met162–Tyr181) of 20, 25 and 20 amino acid residues, respectively. However, the TM2 and TM3 were not aligned at the TM domains of the template. Thus, some manual adjustment of the TM2 and TM3 was performed to prepare the alignment input for the MODELLER. As the three dimensional (3D) structure of the template had four TM domains, only three TM3 domains were considered for modelling the Pseudomonas sp. AMS8 Δ9-fatty acid desaturase (predicted to have only three TM domains) using MODELLER (Fig. 2B). The structure with the lowest DOPE score was assessed and improved after energy minimization and subsequently used for further analyses.

Figure 2 The TM topology as derived from CCTOP prediction (A). Superimposed three dimensional model structures of Pseudomonas sp. AMS8 Δ9-fatty acid desaturase (green) and the template from the crystal structure of mouse stearoyl-coenzyme A desaturase (4YMK) shown in grey (B).

The Zn ions found in the crystal structure of the mouse desaturase are shown as purple spheres. The approximate position of the bilayer is indicated by the two black lines.

Quality verification of the predicted structure

Quality of protein models are verified using various programmes such as ERRAT and Ramachandran plot which are freely available online servers (Colovos & Yeates, 1993). In this study, the predicted structure that has gone through energy minimization was verified for correctness using the ERRAT and Ramachandran plot and labelled as AMX8-em (shown in the supplementary files). The ERRAT programme showed an overall quality value of 65.021% for this structure (Fig. 3). In general, at lower resolution, more than 92% of the surveyed structures had more than 80% of their residues outside the 95% exclusion zone (Colovos & Yeates, 1993). RAMPAGE programme is used to check the overall sterio-chemical quality, local and residue-by-residue reliability usually shown on a Ramachandran plot. The programme shows the sterio-chemistry of the main-chain torsion angles Phi, Psi (φ, ψ) of a good protein model. The Ramachandran plot displays the polypeptide chain of a protein structure using the φ, ψ angles pair (Laskowski, Moss & Thornton, 1993; Mahgoub & Bolad, 2013; Ramachandran, Ramakrishnan & Sasisekharan, 1963). Figure 4 and Table 1 indicate that up to 91.6% of the residues fall within the most favoured regions, 6.8% in the allowed regions whereas only 1.6% residues are in the outlier regions, further confirming that the predicted model is of good quality.

Figure 3 Quality verification plot of the energy minimized model of the Δ9-fatty acid desaturase performed using ERRAT.

The two lines drawn on the error axis show the confidence with which it is possible to reject regions that exceed that error value. In good high resolution structures, 95% or more of the amino acids lie below the 95% threshold whereas in lower resolution (2.5–3 Å) structures around 91% of the amino acids lie below that threshold.

Figure 4 Ramachandran plot of Pseudomonas sp. AMS8 Δ9-fatty acid desaturase model generated by RAMPAGE server.

Table 1 Details of Ramachandran plot after energy minimization.

Plot statistics	% after energy minimization	
Residues in the most favoured regions	91.6	
Residues in allowed regions	6.8	
Residues in the outlier region	1.6	

Catalytic site of the predicted structure

Membrane-bound desaturases share an exceptional structural resemblance and a wide range of functionality. Three conserved-histidine boxes that are common to all classes of these enzymes function in binding two irons at the catalytic centre. The structural similarity has given an insight into their structure-function relationships (Meesapyodsuk et al., 2007). The predicted structure of Pseudomonas sp. AMS8 Δ9-fatty acid desaturase contains the three conserved-histidine boxes consisting of eight histidine residues at positions 1 (His34, His39), 2 (His71, His74, His75) and 3 (His206, His209, His210) from N to C-terminus of the enzyme shown in Fig. 5A and analysed in Fig. 5B. The conserved-histidine motifs are consistent with those observed during the multiple sequences alignments of the target sequence with the template corresponding to the already established catalytic centre of membrane-bound desaturases. The role of the eight histidine residues in the conserved histidine-rich motifs has been demonstrated through site-directed mutagenesis of rat stearoyl-CoA Δ9-desaturase whereas those residues flanking the conserved region have critical catalytic properties in plant FAD2 desaturases and related enzymes (Broadwater, Whittle & Shanklin, 2002; Meesapyodsuk et al., 2007; Shanklin, Whittle & Fox, 1994).

Figure 5 Analysis of Pseudomonas sp. AMS8 Δ9-fatty acid desaturase model showing the overall cartoon representation of the structure, the transmembrane domains are labelled TM1, TM2 and TM3.

The conserved histidine motifs are shown in magenta (A) and the bottom view of the putative catalytic-site residues with the Histidine residues shown in stick representation and the conserved histidine motifs are labelled as 1, 2, and 3 (B).

Docking studies

The membrane-bound Δ9-fatty acid desaturase uses activated oxygen molecule to create double bond between C-H bonds of saturated substrates. The enzyme particularly introduces a double bond at Δ9-position of saturated palmitic and stearic acids to produce palmitoleic and oleic acids, respectively, serving as the fundamental substrates for phospholipids construction and other complex lipid molecules (Castro et al., 2011).

To investigate substrate specificity of Pseudomonas sp. AMS8 Δ9-fatty acid desaturase, docking studies of palmitic acid onto the modeled structure and the template were performed using Autodock software. Blind docking of palmitate and the template was first performed which showed that the palmitate was docked on the template at a site different from the vicinity of the template catalytic site observed for its native ligand. Similarly, for the model structure of desaturase from Pseudomonas sp. AMS8, the docked conformation with lowest docking energy formed close contact with Thr4, Trp167, Val171, Gly170, Leu175, Ala63, Cys96, Gly174, Tyr95, Met141, Ile144, and Ile140 outside the catalytic site. This is expected as the COACH predicted multiple binding sites on the protein.

It is known that the potential catalytic and binding sites for palmitate are close to the His conserved motif. Therefore, specific docking was performed with grid box which covers the histidine residues of the motifs. The docking simulation which produced the lowest binding energy, −6.8 kcal/mol is depicted in Fig. 6A. A number of residues were found in close contact with the docked palmitic acid namely, Ile26, Tyr95, Val179, Gly180, Pro64, Glu203, His34, His206, His71, Arg182, Thr85, Lys98 and His177 (Fig. 6B). Interestingly, among the binding residues are His34 and His71 and His206 from the first, second, and third conserved histidine motif of the enzyme, respectively. The ligand formed two hydrogen bonds with Lys98 and His177. These suggest that the docked substrate was very close to the enzyme catalytic site and the conserved-histidine residues holding the metal ions of membrane-bound desaturases which are known to play a key role for the enzymes catalytic activity (Shanklin et al., 2009).

Figure 6 Docking studies of the 3D structure of palmitic acid onto the predicted model of the Δ9-fatty acid desaturase.

The protein-ligand interactions are shown in surface (A) and the residues involved in binding the ligand (B) analysed using PyMOL software. Two potential hydrogen bonds predicted between Lys98 (K98) and His177 (H177) and palmitate are shown as dotted black lines.

To ascertain the role of the metal irons on the membrane-bound fatty acid desaturases, docking simulations were performed on both the model (Fig. 7) and template (Fig. 8) structures in the presence and absence of metal irons at their catalytic sites. Using the Autodock, different docking scores were predicted and summarized in Table 2. Lowest docking energy (−6.81) was observed when the palmitate was docked onto the model in the presence of metal irons, which suggests a more favourable configuration (Fig. 9). However, the control docking simulations performed on the template with the palmitate in the presence and absence of the iron metals showed much higher docking energies of −6.21 and 5.82, respectively, which indicate that the palmitate was unfavourably bound to the mouse desaturase enzyme catalytic site even in the presence of the metal irons.

Figure 7 Superpose of ligand binding mode on the target site of the model structure.

The palmitate is shown using stick representation in the presence of di-iron (C atom in green) and in the absence of the di-iron (C atom in orange) with O and H atoms in red and white, respectively. The di-irons are shown in purple spheres. Positions of the irons were estimated based on their locations in the template structure (PDB ID: 4YMK).

Figure 8 Superpose of the palmitate binding mode on the active site of the template (PDB ID: 4YMK) in the presence (red) and absence of di-iron (blue).

The ligands are shown in stick representation and the di-iron as purple spheres.

Figure 9 The docked palmitate on the active site of the model structure in the presence of di-iron.

The ligand is shown as spheres with C, O and H atoms in green, red and white, respectively. The di-iron is shown as purple spheres.

Table 2 Summary of docking simulations performed using Autodock.

System	Autodock score (a)	Amino acid residues involved in hydrogen bond interactions	
Model + Palmitate + di-iron (default charge, 0)	−6.81	Arg65, His71	
Model + Palmitate + di-iron (charge = + 1)	−6.48	Arg65, His71	
Model + Palmitate + di-iron (charge = + 2)	−6.74	Arg65, His71	
Model + Palmitate	−6.26	Arg65, His71	
4YMK+ Palmitate + di-iron	−6.27	Asn144	
4YMK+ Palmitate	−5.82	Trp258	
Notes.

a The scores were selected based on the highest conformation observed in the docking simulations.

Simulation of the predicted model of the Δ9-fatty acid desaturase in membrane

In order to further assess the Δ9-fatty acid desaturase model, the protein was embedded within a POPC bilayer and simulated for 50 ns. The approximate location of the bilayer was predicted based on the position of bilayer of the template structure. During equilibration, the protein movement was restrained for 1 ns to allow lipid molecules to equilibrate around the protein. The initial structure of the protein inside the POPC bilayer and its final structure at the end of the MD simulation are shown in Fig. 10.

Figure 10 A snapshot of the atomistic MD simulation of the Δ9-fatty acid desaturase in POPC lipid bilayer at t = 50 ns.

The protein is shown in cartoon representation in grey with the TM domains coloured as blue, red, and yellow for TM1, TM2 and TM3, respectively. The conserved histidine motifs are highlighted in magenta. The POPC lipid molecules are shown in line representation with carbon, nitrogen and oxygen atoms in cyan, blue and red, respectively. Water molecules are hidden for clarity.

Figure 11 Total RMSD of the Δ9-fatty acid desaturase simulated in POPC lipid bilayer and RMSD of its different regions simulated in a POPC lipid bilayer.

Figure 12 ERRAT analyses of the structures at 25 ns (A), 30 ns (B) and 50 ns (C) which produced overall quality scores of 90.81, 93.33 and 84.74%, respectively.

In the membrane, the protein root mean square deviation (RMSD) was calculated to check the overall structure stability (Fig. 11). The RMSD underwent major changes in the first 5 ns and became more stable after 20 ns. To ascertain the real cause of the significant changes observed during the MD simulation, a more fine-grained analysis of the RMSD of the entire protein and its different regions (specifically the TM domains 1 to 3, and the C-terminal region following the last residue of TM3 (182 to 252) of the model structure) along the simulation was carried out. The RMSD established an overall stable protein, which is well equilibrated in the POPC bilayer after the 20 ns. The pdb coordinates of the protein analysed at 25, 30 and 50 ns were extracted from the MD trajectory and performed ERRAT analyses which showed overall quality scores of 90.81, 93.33 and 84.74%, respectively (Fig. 12).

Discussion

Fatty acid desaturase enzymes are involved in unsaturated fatty acid synthesis through desaturation reactions and usually have specificity for double bond insertion along the saturated acyl chains (Los & Murata, 1998; Wang et al., 2013). Membrane-bound fatty acid desaturases perform dehydrogenation reactions of fatty acyl chains that are non-heme di-iron and oxygen-dependent (Meesapyodsuk et al., 2007). Contrary to soluble fatty acid desaturases which have been extensively studied, structural information about the membrane-bound fatty acid desaturases is very limited. Membrane-bound fatty acid desaturases have been isolated and characterised from bacteria (Garba et al., 2016a; Li et al., 2008), fungi (Chen et al., 2013), plants (Gao et al., 2014; García-Maroto et al., 2002) and animals (Bai et al., 2015; Wang et al., 2013). However, the only membrane-bound fatty acid desaturases that have been crystallised so far were reported from animals such as mouse stearoyl-CoA desaturase (Bai et al., 2015) and human stearoyl-CoA desaturase (Wang et al., 2013). Both the primary sequence and the modelled structure of the Pseudomonas sp. AMS8 Δ9-fatty acid desaturase revealed the presence of three conserved-histidne residues at positions 34–39, 71–75 and 206–210, which are typical for all membrane-bound desaturases and play a vital role for the enzymes catalytic activity (Shanklin et al., 2009) as shown in Figs. 1 and 5, respectively. Moreover, multiple sequences alignments of the template and the target showed an extension of amino acids (Val 253 to Ala394) at the C-terminal tail of the target which is completely not observed in the template. Therefore, only residues 1 to 252 were included in the model structure. However, BlastP at NCBI showed that, the extension shares 93% identity to both aminotransferases and acyl-CoA desaturases of many Pseudomonas species.

Contrary to the crystallised structures of other membrane-bound desaturases such as the mouse stearoyl-CoA desaturase (Bai et al., 2015) and the human integral membrane stearoyl-CoA desaturase (Wang et al., 2013), which both had four (4) transmembrane domains, the modeled structure of the Pseudomonas sp. AMS8 Δ9-fatty acid desaturase has only three (3) transmembrane domains (Fig. 2) which are thought sufficient enough to span the membrane bilayer twice with both protein termini facing the cytosol. Although to the best of our knowledge, there was no report on the biding residues for palmitic acid from membrane-bound Δ9-fatty acid desaturase, Ile26, Tyr95, Val179, Gly180, Pro64, Glu203, His34, His206, His71, Arg182, Thr85, Lys98 and His177 were found to bind this substrate (Fig. 6). Among these residues, Ile, Val, Gly, and Arg are comparable to binding residues for stearoyl-CoA by the crystallised structure of a human stearoyl-Coenzyme A desaturase (Wang et al., 2013). Similarly, Arg, Ile, Val, Gly are comparable to some binding residues for stearoyl-CoA of a mammalian steayol-CoA desaturase (Bai et al., 2015).

Supplemental Information

Data S1 Model and docking structures

Click here for additional data file.

Supplemental Information 1 Model Quality verification

Click here for additional data file.

Additional Information and Declarations

Competing Interests

Author Contributions

Data Availability

The authors declare there are no competing interests.

Lawal Garba and Khairul Bariyyah Abd Halim performed the experiments, prepared figures and/or tables.

Mohamad Ariff Mohamad Yussoff and Siti Nor Hasmah Ishak performed the experiments.

Mohd Shukuri Mohamad Ali analyzed the data, authored or reviewed drafts of the paper, co-supervisor.

Siti Nurbaya Oslan contributed reagents/materials/analysis tools, authored or reviewed drafts of the paper, co-supervisor.

Raja Noor Zaliha Raja Abd. Rahman conceived and designed the experiments, authored or reviewed drafts of the paper, chairman supervisory comittee.

The following information was supplied regarding data availability:

The homology modeling experiments, docking studies of a fatty acid desaturase and all the necessary accession numbers are uploaded as a Supplemental File.

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
