# Peer review of "Homology modeling and docking studies of a Δ9-fatty acid desaturase from a Cold-tolerant Pseudomonas sp. AMS8"

_PeerJ, doi:10.7717/peerj.4347_

## Round 0.1 · original submission · Major Revisions

In addition to the issues pointed out by our reviewers, I think that the low sequence identity of your templates (around 25%) makes the straightforward homology modeling approach very dubious. Besides the changes suggested by the referees, you should endeavor to show that your model is stable by performing molecular dynamics simulations in (ideally) an embedded membrane environment.

Reviewer 1 ·

Basic reporting

In this manuscript, Garba et al report a structural model for a bacterial fatty acid desaturate, as well as docking studies to its palmitic acid substrate. Specifically, they identify two mammalian desaturases of known structures, which they use as template to model the bacterial homologue. They then refine their model my energy minimization, and verify its geometry. Finally, they use this model for docking studies with Palmitic acid.
Predicting the structure of membrane-embedded enzymes is clearly an important and challenging problem. In particular, bacterial lipid modifiers such as desaturates are potential antibiotic targets, and therefore understanding the molecular details of their mechanism is highly relevant, However, here the authors employ a very simplistic methodology based on automated servers. Furthermore, the complete lack of biological insights from this bioinformatics analysis renders this study rather moot. Finally, the figures are very poorly rendered, and largely consist of screenshots from the various programs and/or servers used.

Experimental design

- The abstract and introduction mainly discuss the difficulty of protein structure determination and the advantages of homology modeling. This is widely known, and has been reviewed in many articles. The authors should rather focus on their target enzyme: what is known about desaturases mechanism, how the bacterial enzyme differs from other desaturases, and how a structural model could explain its difference in mechanism.

- While YASARA is a commonly used program for homology modeling, it has not been developed specifically for membrane proteins, which represent a specific challenge. Using programs specifically designed for membrane proteins, such as Rosetta, Medeller and/or MEMOIR would be better suited for this study. Similarly, it is not clear if Zn ions and lipids are present in the docking experiments reported here. These would be essential to make sure that the results are accurate.

- In the sequence alignment shown in figure 2, it seems that the target enzyme possesses a ~120 amino-acid extension at the C-terminus. What is the predicted structure of this region, and does it have sequence homology to any other proteins? Was this domain included in the structural model?

Validity of the findings

- The discussion barely shows any biological insights. The authors mention the presence of iron in some membrane-bound desaturates, but the two structures used as template for the modeling clearly have zinc, and therefore the results of this study cannot be used to argue for a difference in ion coordination.

- For such study, the structural model and ligand pdb files should be included in the supplementary material, so that it can be used by readers.

Additional comments

- Figures 1 and 4 seem to be simple screen shots of a server results page, and should be re-done, including labels specific to this work. Figure 7 is unreadable and needs to be re-done. Figures 3 and 6 should be improved, with better color coding, labeling important residues/domains etc. For example, the side-chains in Figure 3 are superfluous, and so are the hydrogen atoms on figure 6.

Reviewer 2 ·

Basic reporting

The approach is good but outcome is not supported by the strong in-silco methods as well as experimental validation therefore real result might be different from what is presented here.

Experimental design

Experimental design is poor. Docking result need to be validated with molecular mechanics methods and molecular dynamics simulation.

Validity of the findings

In my view, finding can not be validated from result provided.

Additional comments

The approach have novelty but lack design and experimental validation therefore in my view, the present work is good start but author have to perform high throughput computational methods to reach at conclusion.

Reviewer 3 ·

Basic reporting

The introduction needs more detail. I suggest that you improve the description abot the enzymatic mechanism of Δ9-fatty acid desaturases.

Please, provide newer references about molecular docking calculations (line 85).

I commend you to provide a description for the initials GCMS (analysis) on line 138.

Figure 1 does not present clear relevance, thus, I suggest you to remove it and solely keep the mention on the text.

Figure 2 presents high relevance on your work and I commend you to improve it, since the given format is the automatically provided by the Clustal software.

The English language is well written, but I recommend the revision of caption 3 (figure 3).

Figure 6 highlights important findings of you work, please, provide a more ilustrative structure, using different colors for secondary structures and for Zn ions.

Figure 7 should also be improved for a better understand of your data. Sticks over transparent cartoon (or only sticks) in a white background, instead of lonely lines in a black background, are more clear representations.

Experimental design

I commend the authors to provide a better description of structure prediction methodology.

I encourage you to provide the specific parameters applied for docking calculation in order assure the results reproducibility.

Since most results are based on docking calculations, I strongly commend the authors to perform redocking calculations (using the mentioned crystal structures) to validate the docking system, in view of the high flexibility of the analysed ligand.

Validity of the findings

The reported findings need to be well discussed to highlight its relevance. Discussion related to the findings are too brief, and the modeled structure is not properly compared to the available crystal structures, especially about ligand binding. Please, provide a extended discussion comparing the molecular basis for ligand binding among the modeled and the crystal structures.

---

## Round 0.2 · Minor Revisions

Your manuscript has been much improved. Please address the remaining issues mentioned by reviewer #1, as well as the following points:

- Like reviewer 1, I think that a more fine-grained analysis of the RMSD of different regions of the protein along the simulaiton is needed, to ascertain whether that changes you see reflect a "unraveling"/instability of the model or are simply due to movements of one domain relative to the rest of the protein. An ERRAT analysis of the structure at different points of the simulation might also be useful.

- The lines where you mention the failure of Autodock to properly dock the substrate in the experimental pose might be written more clearly. I would like to know how the docking energies of this "wrong" pose compare to the Autodock energies of the correct pose.

-The reference to ERRAT is wrong: Maghoub and Bolad (2013) is simply one paper which uses ERRAT. Colovos and Yeats (1993) is the correct reference.

- The explanation of the ERRAT results is also deficient. You state "In general, high resolution structures generate quality values that fall around 95% or higher whereas lower resolution structures produced an average quality factor that is around 91% and the error function is statistically determined on the basis of non-bound atom to atom interactions in the target structure (Colovos and Yeates, 1993)" but I think you mean "In general, 95 % of the residues in high resolution structures lie below the 99% exclusion zone, whereas for lower resolution structures around 91% of the residues lie outside that zone [etc...]" A reference for these assertions is needed, as Colovos and Yeats (p.1516) state something a little different: that at lower resolution more than 92% surveyed structures had more than 80% of their residues outside the 95% exclusion zone.

Reviewer 1 ·

Basic reporting

The revied version of the manuscript is significantly improved, with clearer biological insights and much better figures.

There are two labels in figure 2A that seem to show the protein termini, but are clearly out of place. This should be fixed. In addition, the few lies at the top of figure 3 are not required and should be cropped out.

Experimental design

- The modeling procedure is now well supported, except for the fact that two models were generated (AMX8-em1 and AMX8-em2) but it's not clear what is the difference between them. Were they generated using the two different templates?

- For docking experiments, a few additional controls are required: With and without iron for the model, as well as a control docking in one of the experimentally-determined stearoyl-coA desaturase structure.

Validity of the findings

The overall conclusions are supported, provided the above controls are performed.

A few concept could be explored further:
- The MD simulations show significant changes over time, which part(s) of the protein is(are) affected?
- The authors mention that a three-TM desaturase is unusual. Is it unique to this specific enzyme, or more broadly applicable to a certain sub-family?

---

## Round 0.3 · Minor Revisions

Thank you for performing the requested revisions. I am afraid that some issues still must be addressed before I can accept your paper:

Protein RMSD in Figure 12 is approximately 50% higher than the "total RMSD" in figure 11 throughout the simulation, although the captions imply that the same measurements are being plotted in both graphs. Please check and correct the figures (or captions) accordingly.

Caption to figure 3 has not been fully corrected to the proper description of the meaning of ERRAT scores. The final sentence should read "In good high resolution structures, 95% or more of the aminoacids lie below the 95% threshold whereas in lower resolution (2.5- 3Å) structures around 91% of the aminoacids lie below that threshold" instead of " Good high resolution structures generally produce values around 95% or higher whereas lower resolution (2.5- -3Å) have an average overall quality factor around 91%."

in table 2, you state that di-iron charges are 0, 1 or 2. Normal oxidation numbers of Fe in proteins are, however, +2 or +3. Did you really use individual 0 and 1 charges or are those values the charges of the Fe complex (i.e. summing the charges of the irons to the charges of the complexing aminoacids)? Please clarify or correct, as necessary.

---

## Round 0.4 · accepted · Accept

I am satisfied with your revisions.